# AN ENSEMBLE VIEW ON MIXUP

## ABSTRACT

Deep ensembles are widely used to improve the generalization, calibration, uncertainty estimates and adversarial robustness of neural networks. In parallel, the data augmentation technique of *mixup* has grown popular for the very same reasons. Could these two techniques be related? This work suggests that both implement a similar inductive bias to "linearize" decision boundaries. We show how to obtain diverse predictions from a single mixup machine by interpolating a test instance with multiple reference points. These "mixup ensembles" are cheap: one needs to train and store one single model, as opposed to the $K$ independent members forming a deep ensemble. Motivated by the limitations of ensembles to model uncertainty far away from the training data, we propose a variant of mixup that builds augmented examples using both random interpolations and *extrapolations* of examples. We evaluate the efficacy of our proposed methods across a variety of in-domain and out-domain metrics on the CIFAR-10 and CIFAR-10-NEG datasets.

## 1 INTRODUCTION

In an unprecedented *tour de force*, deep learning has surpassed human performance on a variety of applications such as image classification, speech recognition, natural language processing, and game playing (LeCun et al., 2015; Silver et al., 2016; Goodfellow et al., 2016). While this is nothing short of an incredible achievement, there is a growing interest in the research community to evaluate machine learning models beyond average accuracy (Stock and Cisse, 2018). This is because, regardless of their impressive accuracy in-domain, deep learning models crumble under distribution shifts (Arjovsky et al., 2020; Gulrajani and Lopez-Paz, 2020), have limited ability to say "I don't know" (Tagasovska and Lopez-Paz, 2019; Belghazi and Lopez-Paz, 2021), and are fooled by "adversarial" perturbations imperceptible to humans (Goodfellow et al., 2015). In sum, we have built machines extremely predictive under the conditions they are trained on, but unreliable under changing circumstances.

Deep ensembles (Lakshminarayanan et al., 2016) are a popular technique to improve the generalization (Wilson and Izmailov, 2022), calibration (Guo et al., 2017), uncertainty estimates (Belghazi and Lopez-Paz, 2021; Ovadia et al., 2019), and adversarial robustness (Pang et al., 2019; Abbasi et al., 2020; Adam and Speciel, 2020) of deep neural network models. To construct a deep ensemble, practitioners (i) initialize a collection of $K$ neural networks at random, called ensemble members; (ii) train them all to convergence on the same data; (iii) use as prediction rule the average of the outputs of the $K$ trained neural networks. As such, deep ensembles are $K$ times more expensive than single neural network models, both at training and evaluation time. Moreover, while the uncertainty estimates provided by deep ensembles result in better-calibrated decision boundaries in-between classes, how to signal low confidence far away from training data remains an open question (Amersfoort et al., 2020).

On a separate thread of research, the regularization technique *mixup* (Zhang et al., 2018) has raised in popularity for exhibiting the same benefits as deep ensembles. In a nutshell, mixup trains neural network classifiers on convex combinations of random pairs of examples, and the corresponding label mixtures. More specifically, deep neural networks trained with mixup achieve better generalization (Chun et al., 2020; Zhang et al., 2021a), calibration (Thulasidasan et al., 2020; Zhang et al., 2021b), uncertainty estimates (Lee et al., 2021), and adversarial robustness (Pang et al., 2020; Archambault et al., 2019; Zhang et al., 2021a; Lamb et al., 2021).

In this work, we take the parallels above as hints to consider mixup as an implicit ensemble method. Our main intuition is summarized in Figure 1. In the zero-training-error case, the members of a deep

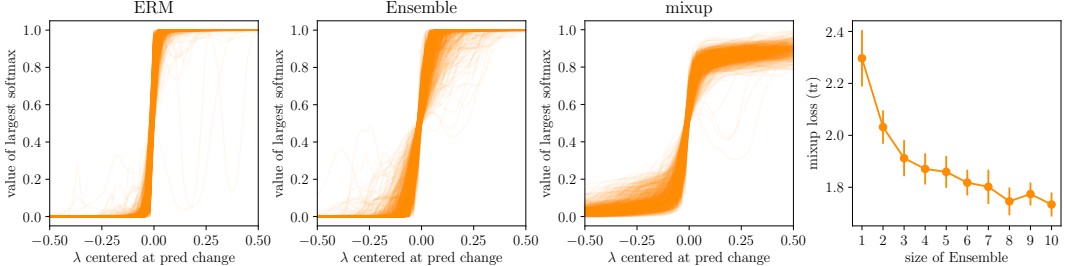

Figure 1: Deep ensembles and mixup exhibit smoother decision boundaries than ERM (first three panels). On the one hand, ensembles "linearize" their decision boundaries by averaging zero-training-error predictors that fluctuate randomly in-between training examples. On the other hand, mixup optimizes for this linear behavior explicitly during training. As we add more members to a deep ensemble, the resulting predictor exhibits a lower mixup loss, while it has never been trained to minimize such statistic (fourth panel). Do ensembles and mixup improve generalization because of the same inductive bias?

ensemble fluctuate randomly in-between training examples, because each member is initialized at random. After ensembling, these random fluctuations cancel each other, smoothening (linearizing) decision boundaries, often improving generalization. This is the very inductive bias directly optimized by mixup! In fact, as we increase the number of members in a deep ensemble, the resulting predictor minimizes the mixup loss implicitly, as each of the members is an ERM model.

**Contributions**   If our intuitions are correct, how could we leverage a single mixup machine like an ensemble? We propose a recipe that interpolates single test instances with multiple "reference" examples, providing a multiplicity of predictions (Section 3). mixup ensembles are $K$ times cheaper than deep ensembles, as only one machine is trained and deployed. We also discuss the limitations of deep ensembles when it comes to detecting out-of-distribution examples that lay far away from the training data. Motivated by this shortcoming, we propose Ex-mixup, a variant of mixup that constructs augmented points allowing both the interpolation and *extrapolation* of pairs of training examples (Section 4). We conclude with a series of numerical experiments to verify the efficacy of the proposed methods (Section 5) and some concluding thoughts (Section 6).

## 2   BASICS ON DEEP ENSEMBLES

Let us start our exposition by providing the necessary background about deep ensembles (Lakshmi-narayanan et al., 2016). Constructing a deep ensemble of neural networks (Lakshminarayanan et al., 2016) is as follows:

1. Instantiate $K$ neural networks of equal architecture at different random initializations $\{\theta_1, \ldots, \theta_K\}$. These neural networks, denoted by $\{f_{\theta_1}, \ldots, f_{\theta_K}\}$, are referred to as the *members* of the ensemble.

2. Train the $K$ neural networks via Empirical Risk Minimization (ERM) on the training data.

3. Deploy as the final prediction rule the average of the $K$ neural networks $f(x) = \frac{1}{K}\sum_{k=1}^{K} f_{\theta_k}(x)$. Use "one minus the largest softmax score" $(1 - \max_c f(x)_c)$ as a measure of predictive uncertainty (Hendrycks and Gimpel, 2016; Lakshminarayanan et al., 2016).

Without loss of generality the sequel considers classification problems with $C$ classes. Thus, ensemble members—as well as the ensemble itself—output $C$-dimensional softmax vectors, belonging to the $C$-dimensional simplex $\Delta^C$.

When using modern neural network architectures, one can assume that each of the $K$ members in the ensemble achieves zero training error (Zhang et al., 2017). However, since the members are initialized at random, they may disagree in those regions of the input space where the labeling is noisy (high aleatoric uncertainty) or there was lack of training data (high epistemic uncertainty) (Kendall and Gal, 2017; Hüllermeier and Waegeman, 2020; Tagasovska and Lopez-Paz, 2019). Therefore,

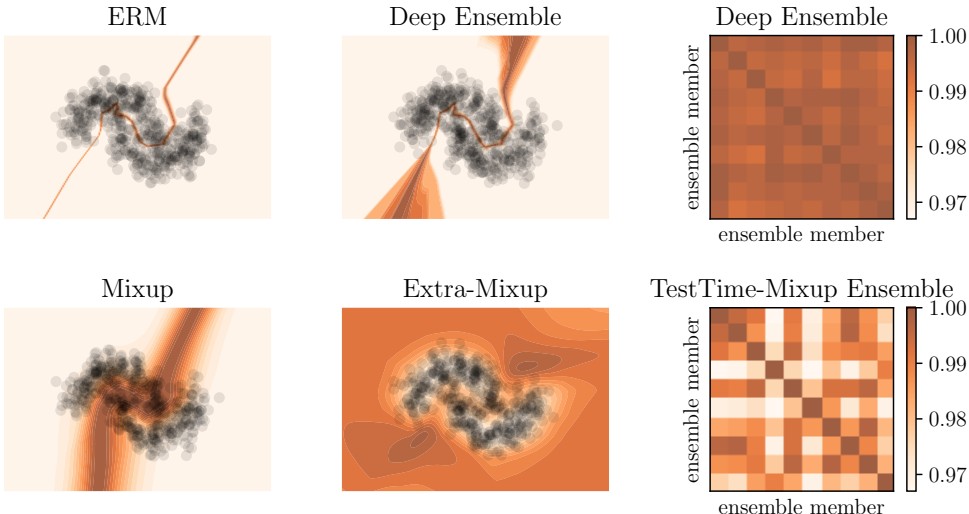

Figure 2: Predictive uncertainty (darker) of the methods discussed in this work. ERM trains a single machine to minimize the prediction error across training data. Deep Ensemble train $K$ ERM models and returns their average as the final prediction. mixup trains an ERM model on random convex interpolations of examples. Ex-mixup trains an ERM on both (i) random convex combinations of examples, and (ii) random extrapolations of examples and the uncertainty label. We observe that ERM provides a sharp decision boundary with no uncertainty awareness. Deep Ensembles and mixup describe uncertainty along the decision boundary. Ex-mixup provides a full account of uncertainty (awareness of being far away of the training data). The ensemble members extracted from a single mixup machine are diverse (in terms of predicted label disagreement).

the ensemble exhibits some degree of uncertainty awareness: when the members of the ensemble disagree, the resulting prediction should be regarded as one with low-confidence (Lakshminarayanan et al., 2016; Ovadia et al., 2019).

It is a classic result that ensembling accurate yet diverse predictors improves generalization error (Hansen and Salamon, 1990; Ueda and Nakano, 1996). However, the ensemble may remain confident in those regions of the space where all the members extrapolate in the same manner. This is illustrated in Figure 2, depicting the classification of two-dimensional moon shapes. While the Deep Ensemble is more uncertain along the decision boundary than a plain ERM predictor, both are as confident as they can be as we move far away from the training data in most directions.

Deep ensembles require training and deploying $K$ neural networks, a computational and storage ask $K$ times exceeding those of a single ERM predictor. This shortcoming, combined with the advantages of using deep ensembles, motivate two recent strands of research, those of MIMO networks (Havasi et al., 2021) and model soups (Wortsman et al., 2022) networks.

On the one hand, MIMO networks train a single classifier to jointly predict $K$ targets from $K$ inputs. For instance, if $K = 3$ the implemented classifier $y_1, y_2, y_3 = f(x_1, x_2, x_3)$ accepts three instances once, and predicts the corresponding three labels jointly. Architecturally, MIMO networks require $K$ input and $K$ output linear layers. Thus, the number of total parameters in a MIMO network increases with $K$—for instance, a $48\%$ increase for $K = 3$ in a ResNet18 ImageNet classifier. When deployed, MIMO networks can provide $K$ predictions about a single test instance, which compete in diversity to those provided by deep ensembles (Havasi et al., 2021).

On the other hand, model soups train $K$ independent models $\{f_{\theta_1}, \ldots, f_{\theta_K}\}$ of equal architecture but different non-architectural hyper-parameters (such as random initialization, learning rate, weight decay). After these $K$ models are trained, we construct a single model, the "soup" $f = \frac{1}{K} \sum_{k=1}^{K} f_{\theta_k}$, by averaging the optimized parameters of the $K$ trained models. Then, building a model soup requires

Table 1: Summary of different learning methods, together with the number of machines necessary to be trained (#tr) and deployed (#te), number of predictions available per test instance (#preds), and level of uncertainty awareness. For each method we list the formula to obtain the $k$-th prediction $p_k$ about a single test instance $x$.

| method | prediction $p_k =$ | #tr | #te | #preds | uncertainty |
|---|---|---|---|---|---|
| ERM | $f_\theta(x)$ | 1 | 1 | 1 | no |
| Mixup | idem | 1 | 1 | 1 | in-domain |
| Extra-Mixup | idem | 1 | 1 | 1 | (in+out)-domain |
| TestTime-Mixup | $(f_\theta(\lambda_k x + (1-\lambda_k)x_k) - (1-\lambda_k)f_\theta(x_k))\lambda_k^{-1}$ | 1 | 1 | $K$ | in-domain |
| Deep Ensemble | $f_{\theta_k}(x)$ | $K$ | $K$ | $K$ | in-domain |
| MIMO | $f_\theta(x, \ldots, x)_k$ | $O(K)$ | $O(K)$ | $K$ | in-domain |
| MC-Dropout | $f_{\theta \odot M_k}(x)$ | 1 | 1 | $K$ | in-domain |
| Soup | $f_{\frac{1}{k}\sum_{k=1}^{K}\theta_k}(x)$ | $K$ | 1 | 1 | no |

training $K$ networks and deploying one. However, a model soup can only provide a single prediction per test instance. Table 1 summarizes the discussed methods so far.

## 3 EXTRACTING ENSEMBLES FROM SINGLE MIXUP MACHINES

The mixup data augmentation technique (Zhang et al., 2018) trains classifiers on random convex combinations $(\tilde{x}, \tilde{y})$ of training examples:

$$(\tilde{x}, \tilde{y}) = \text{mixup}(x_i, y_i, x_j, y_j, \lambda) := (\lambda \cdot x_i + (1-\lambda) \cdot x_j, \lambda \cdot y_i + (1-\lambda) \cdot y_j).$$

In the previous, $(x_i, y_i)$ and $(x_j, y_j)$ are a random pair of training examples, and $\lambda \sim \text{Beta}(\alpha, \alpha)$ is a random mixing coefficient. The mixup parameter $\alpha$ controls the strength of mixing, recovering the vanilla ERM training as $\alpha \to 0$. Once deployed, the mixup machine is evaluated normally at test instances. Therefore, mixup trains and deploys a single machine, with the capability of providing a single prediction per test instance.

mixup training has gained traction due to improving generalization error (Chun et al., 2020; Zhang et al., 2021a), uncertainty estimates (Thulasidasan et al., 2020; Zhang et al., 2021b; Lee et al., 2021), and adversarial robustness (Pang et al., 2020; Archambault et al., 2019; Zhang et al., 2021a; Lamb et al., 2021). Figure 2 shows how mixup produces a smooth, well-calibrated decision boundary in a two-dimensional classification problem. As stated in the introduction, these are the same suspected reasons behind the success of deep ensembles. Could these two techniques be related?

In the following, we outline a simple procedure to leverage a single mixup machine to produce $K$ predictions about a single test instance.[1] To this end, consider a small collection of $K$ reference points $\{(x_k, y_k)\}_{k=1}^{K}$. These could be training or held-out examples or, when adversarial robustness is a worry, kept secret during deployment. By virtue of the mixup condition, the machine trained with mixup should satisfy $f(\lambda_k x + (1-\lambda_k)x_k) \approx \lambda_k f(x) + (1-\lambda_k)f(x_k)$, where $\lambda_k \sim \text{Beta}(\alpha, \alpha)$ uses the same mixing hyper-parameter $\alpha > 0$ as in training time. By rearranging, we use the reference point $x_k$ as an "anchor" to obtain the prediction $f_k(x)$ about the test instance $x$, namely:

$$f(x) \approx \frac{f(\lambda_k x + (1-\lambda_k)x_k) - (1-\lambda_k)f(x_k)}{\lambda_k} =: f_k(x). \tag{1}$$

Repeating with all reference points $\{x_k\}_{k=1}^{K}$, we are now able to construct an ensemble of predictions $\{f_1(x), \ldots, f_K(x)\}$ about $x$ using a single mixup machine. To produce a valid simplex vector in $\Delta^C$, we perform only softmax operation as the last step in Equation (1). In practice, we choose the reference points by running $k$-kmeans in feature space, and selecting the raw training samples closest to each centroid. We call this a TestTime-mixup (TT-mixup) ensemble of $K$ members.

The mixing coefficients $\lambda_k$ offer an opportunity to trade-off prediction diversity ($\lambda_k \to 0$) and maximum-a-posteriori accuracy ($\lambda_k \to 1$). Figure 2 shows that the member predictions of TT-mixup are diverse, even more so than the ones provided by the expensive Deep ensemble.

---

[1]We found this process was briefly mentioned in an unpublished manuscript (Lee et al., 2021).

**MIMO *versus* mixup**    One can view MIMO (wlog $K = 2$) as a generalization of mixup, where on trains a classifier on "mixed" data $(\tilde{x}, \tilde{y}) = ((x_i, x_j), (y_i, y_j))$. While mixup fixes the construction of mixed data as a random convex combination, MIMO employs the more general concatenation operator to construct augmented examples, delegating their mixing to the first convolutional layer. While MIMO's generality comes at the expense of additional parameters, the retrieval of its member predictions $f_k(x) = f(x, x)_k$ is more transparent. These and other techniques fall under a common framework of "learning from multiple examples".

## 4    WHAT HAPPENS OUT-OF-DISTRIBUTION?

Given the importance of out-of-distribution (ood) performance and uncertainty estimation in our enquire, we cannot help but notice in Figure 2 a worrisome behaviour for ERM, Deep Ensemble, and mixup machines. The machines produced by these three training algorithms are maximally confident about their predictions as we move far away from training data in most directions (the confidence turns white away from the center). This is an niggling limitation in safety-critical applications (Kendall and Gal, 2017), where we would desire that our machine learning systems signal low-confidence and abstain from prediction in those extreme regions of the input space.

This discussion is an instance of striving for machine learning methods able to model both aleatoric and epistemic uncertainty (Tagasovska and Lopez-Paz, 2019). On the one hand, we observe *aleatoric* uncertainty in those regions of the input space where the true labeling mechanism is noisy. In Figure 2, this corresponds to the decision boundary separating the two overlapping classes—no matter how much additional data we collect, predicting close to the boundary is prone to some irreducible uncertainty. On the other hand, we *should* observe high *epistemic* uncertainty in those regions of the space where we have not observed any training data. Therefore, epistemic uncertainty is inversely proportional to the input density of the training distribution. In Figure 2 epistemic uncertainty should increase (color should darken) as we move far away from the center in any direction.

To address this issue, we extend mixup training to construct augmented data not only interpolating, but also extrapolating, random pairs of training examples. More specifically, we consider training our classifier of choice on mixed examples built as:

$$(\tilde{x}, \tilde{y}) = \begin{cases} \text{mixup}(x_i, y_i, x_j, y_j, \lambda) & \text{or} \\ \text{mixup}(x_i, y_i, 2x_i - x_j, u, \lambda) & \text{with equal probability.} \end{cases} \tag{2}$$

Like in mixup, in the previous equation $(x_i, y_i)$ and $(x_j, y_j)$ are a random pair of training examples, $\lambda \sim \text{Beta}(\alpha, \alpha)$ is a random mixing coefficient, and $u = (1/C, \dots, 1/C)$ signals highest uncertainty.

The first case of the augmentation protocol (2) is vanilla mixup, where we interpolate between random pairs of instances and their labels. The second case of (2) allows to extrapolate or "over-shoot" beyond the first example $x_i$ of the selected random pair. When this happens, the accompanying label is an interpolation between the label associated to the first selected example $x_i$ and the highest-uncertainty label $u$. Intuitively, we are providing signal to the machine as to become uncertain outside of the convex hull described by the training data. We call this training scheme *Ex-mixup-v1*. Figure 2 illustrates the impact of Ex-mixup-v1 training on the confidence surface of the resulting classifier. Now, we observe increasing uncertainty anywhere but where the training data lives. This captures both aleatoric uncertainty (in-between classes) and epistemic uncertainty (far away from data).

### 4.1    A SECOND VERSION OF EX-MIXUP

In Ex-mixup-v1, choosing the highest uncertainty label to be $u = (1/C, \dots, 1/C)$ may lead to unpredictable behavior when moving *very* far away from the training data. In particular, since the value $u = (1/C, \dots, 1/C)$ is a non-saturated softmax vector, the output of the network is free to drift to a saturated softmax vector far-away from the training data. To address this issue, our experiments also implement a second version of Ex-mixup, termed Ex-mixup-v2.

Ex-mixup-v2 instantiates an additional, dedicated $(C + 1)$-th uncertainty label. Therefore, the labels of normal training points $y_i$ are extended to be $y_i = (y_i, 0) \in \Delta^{C+1}$, and the highest-uncertainty label is set to $u = (0, \dots, 0, 1) \in \Delta^{C+1}$. Figure 3 shows the different behavior of Ex-mixup-v1 and Ex-mixup-v2 in the two moons problem when moving far away from the training data.

| mixup | Extra-mixup-v1 | Extra-mixup-v2 |

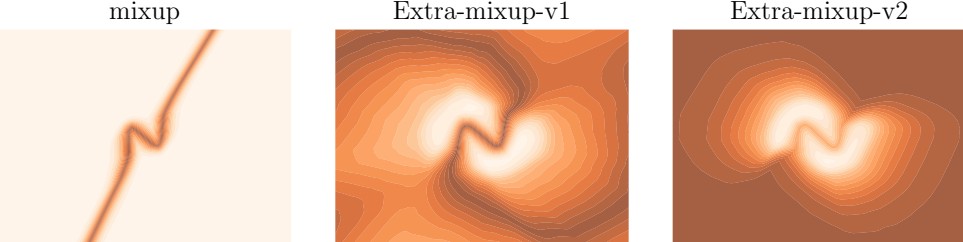

Figure 3: Comparison of uncertainty landscapes (darker is less confident) for a binary classification problem as provided by mixup, Ex-mixup-v1, and Ex-mixup-v2. Mixup provides limited uncertainty awareness, orthogonal to the 1-dimensional manifold described by the decision boundary. Ex-mixup-v1 is more uncertainty-aware, but the use of the softmax non-saturating value $\left(\frac{1}{2}, \frac{1}{2}\right)$ to signal uncertainty results in unpredictable confident far away from the training data. Ex-mixup-v2 fixes this issue by extending the problem to three classes, where high-uncertainty extrapolations are labeled using the softmax saturating value $(0, 0, 1)$.

Lastly, Ex-mixup-v2 allows the separate modeling of aleatoric and epistemic uncertainty. On the one hand, the $(C + 1)$-th softmax score signals epistemic uncertainty. On the other hand, the entropy of the first $C$ logits signals aleatoric uncertainty. When the $(C + 1)$-th score equals one, we are in a situation of maximum epistemic uncertainty and null aleatoric uncertainty. When the $(C + 1)$-th score is zero, there is null epistemic uncertainty, and an aleatoric uncertainty proportional to the entropy of the first $C$ logits.

## 5 EXPERIMENTS

We conduct experiments based on the **CIFAR-10** dataset (Krizhevsky, 2009). In addition to the original training (tr) and testing (te) splits, we add the **CIFAR-10-Neg** dataset (Kalpun et al., 2022) as a third split (ne), containing difficult examples. We divide each split into in-domain data (first five classes, accuracy metrics) and out-domain data (second five classes, ood metrics).

### 5.1 METHODS

We implement six learning techniques:

- **ERM** trains a single machine to minimize prediction error across training examples.
- **TT-Aug** trains a single ERM machine; however, during evaluation time it acts an ensemble by evaluating different random data augmentations of the test instance.
- **Ensemble** trains $K$ ERM predictors, each initialized at random.
- **mixup** trains a single ERM machine on mixed data.
- **TT-mixup** trains a single mixup machine; however, during evaluation time it acts an ensemble by using the interpolation recipe from Section 3, setting $\lambda_k = 0.8$.
- **Ex-mixup (-v1, -v2)** trains a single machine on extrapolated mixed data (Section 4).

All methods are optimized for $500$ epochs, a batch size of $128$, and SGD with a learning rate of $0.1$, a momentum of $0.9$, and a weight decay of $10^{-4}$. The learning rate is divided by ten every $150$ epochs. Each method is trained from $100$ random initializations and data orderings. The number of members in ensembles (as well as the number of evaluations in TT-mixup) is $K = 10$.

### 5.2 METRICS

We measure a variety of metrics to evaluate generalization, robustness, and ensemble diversity.

Table 2: Main results for all methods on CIFAR datasets. Best results at te/ne splits in `orange`. For all methods, we evaluate classification performance (avg acc), adversarial robustness (adv acc), ood detection (OOD acc), loss raw data (loss), violation of the mixup condition (mixup loss), and expected calibration error (ECE). For ensembles, we also report diversity in-domain (diversity) and out-domain (ood diversity).

| method | ERM | Ensemble | mixup | Ex-mixup-v1 | Ex-mixup-v2 | TT-mixup | TT-Aug |
|---|---|---|---|---|---|---|---|
| avg acc (tr) | $1.000 \pm 0.000$ | $1.000 \pm 0.000$ | $1.000 \pm 0.000$ | $0.998 \pm 0.002$ | $0.999 \pm 0.001$ | $1.000 \pm 0.000$ | $1.000 \pm 0.000$ |
| avg acc (te) | $0.944 \pm 0.007$ | $0.958 \pm 0.001$ | $0.955 \pm 0.006$ | $0.951 \pm 0.006$ | $0.951 \pm 0.006$ | $0.959 \pm 0.003$ | $0.951 \pm 0.003$ |
| avg acc (ne) | $0.657 \pm 0.014$ | $0.686 \pm 0.007$ | $0.688 \pm 0.016$ | $0.681 \pm 0.017$ | $0.685 \pm 0.016$ | $0.721 \pm 0.011$ | $0.674 \pm 0.014$ |
| worst acc (tr) | $1.000 \pm 0.000$ | $1.000 \pm 0.000$ | $0.999 \pm 0.001$ | $0.997 \pm 0.003$ | $0.998 \pm 0.002$ | $1.000 \pm 0.000$ | $1.000 \pm 0.000$ |
| worst acc (te) | $0.909 \pm 0.011$ | $0.930 \pm 0.002$ | $0.930 \pm 0.009$ | $0.919 \pm 0.011$ | $0.919 \pm 0.011$ | $0.931 \pm 0.004$ | $0.918 \pm 0.005$ |
| worst acc (ne) | $0.462 \pm 0.040$ | $0.503 \pm 0.018$ | $0.405 \pm 0.040$ | $0.500 \pm 0.045$ | $0.509 \pm 0.044$ | $0.507 \pm 0.033$ | $0.466 \pm 0.047$ |
| adv acc (tr) | $0.374 \pm 0.031$ | $0.528 \pm 0.012$ | $0.691 \pm 0.042$ | $0.604 \pm 0.061$ | $0.562 \pm 0.073$ | $0.579 \pm 0.022$ | $0.461 \pm 0.024$ |
| adv acc (te) | $0.336 \pm 0.028$ | $0.481 \pm 0.018$ | $0.589 \pm 0.037$ | $0.524 \pm 0.048$ | $0.489 \pm 0.062$ | $0.500 \pm 0.018$ | $0.409 \pm 0.021$ |
| adv acc (ne) | $0.069 \pm 0.013$ | $0.141 \pm 0.011$ | $0.191 \pm 0.026$ | $0.150 \pm 0.028$ | $0.138 \pm 0.039$ | $0.141 \pm 0.014$ | $0.102 \pm 0.008$ |
| ood acc (tr) | $0.845 \pm 0.010$ | $0.906 \pm 0.002$ | $0.785 \pm 0.011$ | $0.788 \pm 0.025$ | $0.612 \pm 0.109$ | $0.823 \pm 0.006$ | $0.879 \pm 0.009$ |
| ood acc (te) | $0.761 \pm 0.011$ | $0.791 \pm 0.003$ | $0.731 \pm 0.009$ | $0.732 \pm 0.022$ | $0.596 \pm 0.089$ | $0.757 \pm 0.008$ | $0.783 \pm 0.009$ |
| ood acc (ne) | $0.444 \pm 0.012$ | $0.438 \pm 0.003$ | $0.482 \pm 0.012$ | $0.470 \pm 0.017$ | $0.489 \pm 0.030$ | $0.465 \pm 0.006$ | $0.447 \pm 0.009$ |
| diversity (tr) |  | $0.975 \pm 0.000$ |  |  |  | $0.966 \pm 0.002$ | $0.975 \pm 0.000$ |
| diversity (te) |  | $0.934 \pm 0.002$ |  |  |  | $0.947 \pm 0.002$ | $0.945 \pm 0.002$ |
| diversity (ne) | N/A | $0.795 \pm 0.004$ |  | N/A |  | $0.876 \pm 0.009$ | $0.850 \pm 0.005$ |
| OOD diversity (tr) |  | $0.810 \pm 0.004$ |  |  |  | $0.876 \pm 0.008$ | $0.854 \pm 0.004$ |
| OOD diversity (te) |  | $0.810 \pm 0.004$ |  |  |  | $0.875 \pm 0.007$ | $0.854 \pm 0.003$ |
| OOD diversity (ne) |  | $0.825 \pm 0.003$ |  |  |  | $0.893 \pm 0.007$ | $0.867 \pm 0.004$ |
| data loss (tr) | $0.000 \pm 0.000$ | $0.000 \pm 0.000$ | $0.139 \pm 0.021$ | $0.306 \pm 0.098$ | $0.093 \pm 0.070$ | $0.111 \pm 0.024$ | $0.000 \pm 0.000$ |
| data loss (te) | $0.308 \pm 0.039$ | $0.138 \pm 0.002$ | $0.240 \pm 0.022$ | $0.396 \pm 0.092$ | $0.230 \pm 0.075$ | $0.204 \pm 0.024$ | $0.186 \pm 0.011$ |
| data loss (ne) | $2.146 \pm 0.101$ | $1.101 \pm 0.032$ | $0.848 \pm 0.035$ | $0.914 \pm 0.050$ | $1.063 \pm 0.109$ | $0.768 \pm 0.028$ | $1.405 \pm 0.063$ |
| mixup loss (tr) | $2.226 \pm 0.086$ | $1.708 \pm 0.038$ | $0.661 \pm 0.021$ | $0.799 \pm 0.054$ | $1.021 \pm 0.128$ | $0.712 \pm 0.037$ | $1.837 \pm 0.069$ |
| mixup loss (te) | $2.442 \pm 0.185$ | $1.723 \pm 0.130$ | $0.741 \pm 0.036$ | $0.856 \pm 0.063$ | $1.104 \pm 0.158$ | $0.748 \pm 0.028$ | $2.045 \pm 0.135$ |
| mixup loss (ne) | $2.780 \pm 0.288$ | $1.473 \pm 0.180$ | $0.963 \pm 0.067$ | $1.074 \pm 0.080$ | $1.343 \pm 0.182$ | $0.930 \pm 0.057$ | $1.926 \pm 0.277$ |
| ECE (tr) | $0.000 \pm 0.000$ | $0.000 \pm 0.000$ | $0.127 \pm 0.017$ | $0.257 \pm 0.069$ | $0.084 \pm 0.059$ | $0.103 \pm 0.021$ | $0.000 \pm 0.000$ |
| ECE (te) | $0.045 \pm 0.006$ | $0.010 \pm 0.002$ | $0.106 \pm 0.017$ | $0.228 \pm 0.065$ | $0.066 \pm 0.052$ | $0.094 \pm 0.021$ | $0.020 \pm 0.001$ |
| ECE (ne) | $0.297 \pm 0.013$ | $0.158 \pm 0.007$ | $0.103 \pm 0.022$ | $0.088 \pm 0.025$ | $0.140 \pm 0.038$ | $0.064 \pm 0.018$ | $0.200 \pm 0.014$ |

- **Classification accuracy** (avg acc) percentage of in-domain images correctly classified.
- **Accuracy on adversarial examples** (adv acc) measures the percentage of adversarially corrupted in-domain images that are correctly classified. The adversarial examples are produced using the fast sign gradient method with $\varepsilon = 0.1$ (Goodfellow et al., 2015).
- **Accuracy at ood detection** (ood acc) measures the binary classification accuracy on the task of distinguishing between in-domain and out-domain images. An image is considered out-domain if the entropy of the ensemble exceeds its 95th percentile (as computed across the training data).
- **Cross entropy loss on raw data** (data loss).
- **Mixup violation** (mixup loss) measures the cross-entropy loss on mixup examples.
- **Expected calibration error** (ECE), which is minimized when the classifier is wrong $(100 \times p)\%$ of the times that it predicts a softmax probability of $p$ (Guo et al., 2017).
- **Ensemble diversity** (diversity) measures the average label prediction disagreement between pairs of ensemble members, across in-domain images.
- **Ensemble diversity ood** (*diversity ood*) is analogous to the above, for out-domain images.

## 5.3 RESULTS

Table 2 summarizes our main results. Overall, mixup methods outperform ERM ensembles for most performance metrics, except for **ood acc (te)** and **ECE (te)**. mixup is the overall best alternative when it comes to (average, worst, adversarial) accuracy, with TT-mixup taking the lead on the (ne) split for average and worst accuracy. We observe that the diversity between member predictions is similar for Ensembles and TT-mixup. When it comes to the proposed Ex-mixup alternatives, none of them seem to provide a worthy edge over vanilla or TT-mixup. This may be related to the fact that

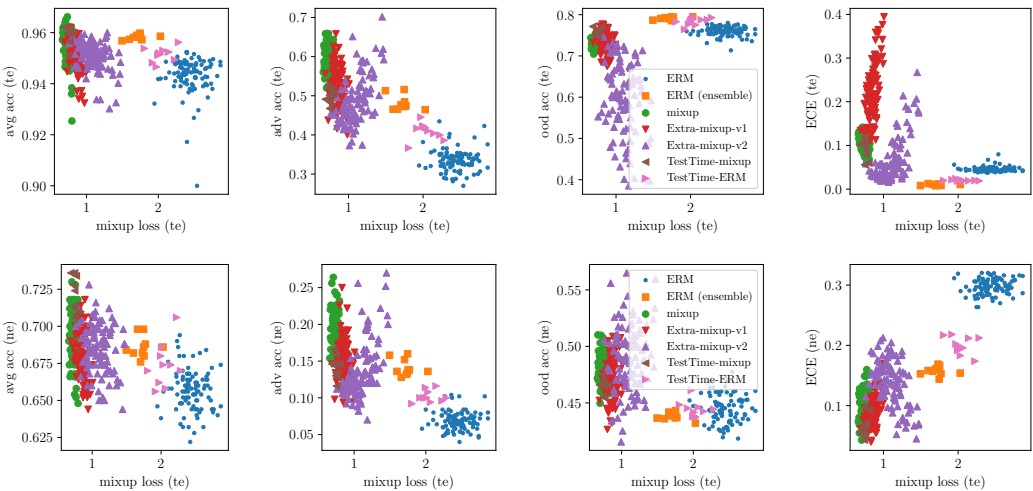

Figure 4: Relationship between mixup loss in-domain (split te) and performance metrics on the te/ne splits. Vanilla and TT-mixup outperform ERM and Ensembles.

"learning in high dimension always amounts to extrapolation" (Balestriero et al., 2021) (in contrast to the optimistic intuitions provided by Figure 3).

One interesting remark is that Ensembles respect the mixup condition much better than single ERMs. While we lack a formal justification, this is likely because the averaging of several functions initialized at random results in linearizing fluctuations in-between training points (where they all match)—the exact criterion explicitly enforced by mixup (recall Figure 1). Thus, it may be that Ensembles and mixup are tightly related in the way they improve in-domain and out-domain performance. In sum, it seems like most benefits from Ensemble methods can be inherited by a single mixup machine used as-is, employing the TT-mixup protocol for improved ood detection and ECE.

**Some ablations** First, increasing the number of ensemble members $k$ is beneficial for both methods and metrics except adversarial accuracy. Second, we observe that $\lambda = 0.8$ is a good trade-off when using the TT-mixup protocol. Third, there is no benefit in adversarial accuracy by keeping the reference points secret (forcing the adversary to use a different set of reference points). Fourth, combining TT-mixup with TT-Aug does not induce a consistent performance gain across metrics. Finally, Figure 4 shows the relationship between mixup loss and four metrics on the te/ne splits. This illustrates the strong relationship between how much the mixup is satisfied and different in-domain and out-domain performance metrics.

**Going beyond simple averaging** While TT-Mixup uses an average of the different members, we also develop variants that uses a weighted sum of the members. For each member, we compute its weight as the sum of its cross-entropy with each of the other members' predictions. Then the member's weight is computed as a softmax over the negative of the summed cross-entropies, with a temperature parameter. As the temperature parameter increases, this converges to a uniform distibution over each member, i.e. taking the average as in TT-Mixup. This can be viewed as using the Maximum Expected Utility principle as in Premachandran et al. (2014), where the member's expected utility (or negative expected cross-entropy) is computed using the other members' predictions. Results are shown in Table 3 for the main accuracy metrics. As expected, a temperature of 10 gets results close to regular TT-Mixup, with a 1.4% improvement on the (ne) split worst accuracy (0.521 versus 0.507 for TestTime Mixup). As the temperature decreases, the strategy works better on the split ood (ne) accuracy, matching mixup performance (0.486 versus 0.482), but at the expense of ood (tr) and (te) accuracies. Nevertheless, low temperatures e.g. a value 0.5 performs on par or outperform Mixup on the main accuracies reported.

Table 3: Weighted sum of member with TT-Mixup, for different temperature parameters. We fix $\lambda = 0.8$ and $k = 20$.

| temperature | 0.1 | 0.5 | 1.0 | 10.0 |
|---|---|---|---|---|
| avg acc (tr) | $1.000 \pm 0.000$ | $1.000 \pm 0.000$ | $1.000 \pm 0.000$ | $1.000 \pm 0.000$ |
| avg acc (te) | $0.958 \pm 0.003$ | $0.958 \pm 0.003$ | $0.958 \pm 0.003$ | $0.958 \pm 0.003$ |
| avg acc (ne) | $0.722 \pm 0.013$ | $0.720 \pm 0.012$ | $0.724 \pm 0.012$ | $0.720 \pm 0.015$ |
| worst acc (tr) | $0.999 \pm 0.000$ | $0.999 \pm 0.000$ | $0.999 \pm 0.000$ | $0.999 \pm 0.000$ |
| worst acc (te) | $0.925 \pm 0.010$ | $0.926 \pm 0.006$ | $0.927 \pm 0.007$ | $0.927 \pm 0.007$ |
| worst acc (ne) | $0.527 \pm 0.040$ | $0.525 \pm 0.028$ | $0.527 \pm 0.042$ | $0.521 \pm 0.038$ |
| ood acc (tr) | $0.768 \pm 0.009$ | $0.787 \pm 0.008$ | $0.792 \pm 0.008$ | $0.815 \pm 0.005$ |
| ood acc (te) | $0.723 \pm 0.009$ | $0.735 \pm 0.007$ | $0.737 \pm 0.007$ | $0.751 \pm 0.006$ |
| ood acc (ne) | $0.486 \pm 0.012$ | $0.482 \pm 0.010$ | $0.477 \pm 0.008$ | $0.466 \pm 0.007$ |

## 6 CONCLUSION

We have offered two ensemble perspectives on mixup. On the one hand, we have introduced TT-mixup, a test-time protocol to extract $K$ diverse predictions about a single test instance from a single mixup machine. This enables the benefits of the widely used deep ensembles at $K$ times less training and deployment cost. On the other hand, we have introduced a Ex-mixup, a novel mixup augmentation protocol to foster better ood awareness far away from the training data. While Ex-mixup does not seem to offer any competitive advantage in our experiments, TT-mixup improves both in-domain and out-domain performance. Interestingly, our implementation of TT-mixup pays no attention to the selection of mixing coefficients (fixed to $\lambda_k = 0.8$). These are all promising venues to improve TT-mixup and bring its performance even closer to the gold standard of deep ensembles. In sum, we hope that this work sheds some light to the inner workings of mixup, and sparks further investigation on how single classifiers can be better leveraged during deployment for increased performance, robustness, and uncertainty awareness.

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
