# OpenReview forum: "An ensemble view on mixup"
_ICLR.cc/2023/Conference — Submitted to ICLR 2023_

### Official Review · Reviewer_1zoz · 2022-10-22

**Confidence:** 4
**Correctness:** 3
**Technical Novelty And Significance:** 2
**Empirical Novelty And Significance:** 3
**Recommendation:** 3

**Clarity, Quality, Novelty And Reproducibility:**

-Quality and Reproducibility: The experiments in this paper use standard procedures for training and testing, and use common metrics for the performances. Besides, most hyperparameters for training are provided, making it possible to reproduce the results. The authors don't seem to provide the network structure they use for the experiments, though.

-Novelty: To the best of my knowledge, the idea of connecting mixup to ensemble is novel and interesting. As for the TT-mixup, as mentioned in the paper, this method is briefly mentioned in an unpublished manuscript, which could decrease the novelty.

-Clarity: This paper is generally well-structured. However, some figures, descriptions, and claims are somewhat ambiguous or lack enough explanation. The organization of the figures could be improved, and there are some typos in this paper. These clarity problems make this paper a bit hard to follow.

Problems related to clarity:
1. In the first 3 plots of Figure 1, what is $\lambda$ (value on $x$ axis)?
2. In Figures 1 and 2, it might be better to separate subfigures of different kinds, e.g., divide them into two groups (a) and (b)
3. For the notations defined at the beginning of Section 2, $\theta_i$ is dependent on time, but the authors are using the same notation for both initialization and learned weights, which might create confusion.
4. After equation (1), the authors claimed that they only perform softmax as the last step in Equation (1), but the definition of $f$ is a probability vector, so equation (1) naturally gives a probability vector. Do the authors use $f$ here to represent the features before the softmax layer?
5. At the beginning of Section 4.1, it might be unclear why the network output is free to shift to saturated softmax vector when the data is far away from training data. Could the authors explain more about this?
6. Figure 4 might seem a bit messy and hard to read since it contains too much information. Besides, it could be better to include the results of temperature equals 0 and infinity, which makes comparison easier.

Typos:

Section 3, 4th line, "In the previous" -> "In the previous equation"

Section 3, 2nd paragraph, "mixup training" -> "Mixup training"

Section 3, 2nd paragraph, "improving generalization error" -> "improving generalization performance"

Page 5, 1st line, "where on trains" seems confusing

Figure 3, caption, "unpredictable confident" -> "unpredictable confidence"

**Strength And Weaknesses:**

-Strengths:

1. The relationship between mixup and ensemble is interesting and novel, and has the potential to explain why mixup works.

2. The authors proposed several versions of their two mixup variants and did ablation studies, which could give more insights into the use of mixup variants.

-Weaknesses:

1. The relationship between ensemble and mixup and the two proposed mixup variants might lack enough justification. I will explain these three points separately below:

1.1 For the relationship between ensemble and mixup, the authors claimed in the paper that both of them encourage the model to have some linear behavior. However, the trained neural networks using these methods are always still highly non-linear except on very simple datasets. Besides, it seems not clear from the paper why this linear behavior is related to the performance (e.g., generalization, robustness) of the models trained.

1.2 For TT-mixup, the reference points are selected by running $K$-means on feature space, and it might be unclear how to choose $K$ and why the cluster centers are good candidates for the "anchors".

1.3 As for Ex-mixup, extrapolation and interpolation are two ways to encourage linearity of model output. However, the authors treat the extrapolation points as out of distribution, which might not be always the case for real datasets. The selection of parameters also seems not well-justified, e.g., $2x_i-x_j$ for the extrapolation and setting the additional uncertainty entry to 1 for extrapolated data points.

2. The experimental results are mixed (different models are good at different tasks and there doesn't seem to be a very clear pattern), and the proposed methods don't show a clear advantage over existing results. Moreover, the results are only on CIFAR-10 and its variants, which might make the conclusions somewhat limited.

3. This paper is not written in a very clear way, and there are some places that are hard to understand. The detail about this will be provided in the Clarity part in the next section.

**Summary Of The Paper:**

This paper relates mixup to ensemble and conjectures that mixup works as an implicit ensemble, which could potentially explain the benefits of mixup, e.g., improving generalization, robustness, etc. Based on this conjecture, the authors proposed two variants of mixup: TT-mixup which improves training and test efficiency upon ensemble, and Ex-mixup to potentially increase OOD accuracy and detection. Experiments were done on CIFAR-10 to evaluate the performances of the proposed methods.

**Summary Of The Review:**

I tend to vote for rejecting this paper. Although the idea of connecting mixup with ensemble is interesting, the justifications for their proposed methods are somewhat unclear, and these methods do not perform consistently better than previous ones in the experiments. The clarity of this paper also needs improvement.

---

> ### Author Response · Authors · 2022-11-11
> **Rebuttal**
>
> Thank you for your review, we are working towards addressing your concerns. About weaknesses:
>
> 1.1. mixup has been transformational in training deep neural network models. The theoretical justification of mixup (based on a second-order expansion of the loss) relates to regularization terms of first and second order derivatives. Empirically, the cited body of work are examples of performance and robustness gains obtained by the enforcement of linearity in-between training examples.
>
> 1.2. We have ran experiments where we choose the K points at random, or at random but one per class, and obtained very similar results. We will include these ablations in the final version.
>
> About clarity:
>
> 1: $\lambda$ is the mixture coefficient between two examples. This allow us to see how does the decision boundary vary when interpolating between two examples. To better compared the sampled decision boundaries, we center them to place their 0.5 $y$-axis point at the middle of the $x$-axis.
>
> 2: Agreed.
>
> 3: Agreed.
>
> 4: That is correct, clarified.
>
> 5: Deep ReLU networks F (ignoring biases for now) are homogeneous functions, that is F(a * x) = a * F(x) for a > 0. Therefore, if we pass to the network 1000 * x, we will get 1000 * F(x). These logits are much larger and will saturate the softmax vector of probabilities. We will explain better this, thank you.
>
> 6: Thank you for this suggestion, we haven't thought of that. We will update as suggested.

---

### Official Review · Reviewer_S9F4 · 2022-10-25

**Confidence:** 3
**Correctness:** 3
**Technical Novelty And Significance:** 3
**Empirical Novelty And Significance:** 2
**Recommendation:** 5

**Clarity, Quality, Novelty And Reproducibility:**

This paper has good clarity and is easy to follow. The idea of exploring the relation between ensemble and mixup is overall novel. The quality depends on whether there is enough evidence to corroborate the promises, which will be my current concern (see the weakness part noted above).

**Strength And Weaknesses:**

Strengths:
- The paper produces quite a few analyses, including (1) drawing attention to the similarity in the effect of ensemble and mixup, (2) how ensemble and mixup do not seem to work well on out-of-distribution inputs (despite later empirically they worked pretty well on the dataset evaluated), (3) how extrapolating mixup generates good looking uncertainty in a decision boundary plot (again despite the empirical experiment).
- The paper is well-written and easy to follow.

Weaknesses:
- The paper seems more like an exploration of mixup methods, covering aspects on its relation with ensemble and ood generation. I did not find affirmative answers on some of the questions I had in mind when reading the paper: (1) how or why mixup is similar to ensemble learning (apart from figure 1, I was hoping for more intuition/proof on why ensemble, something that produces a more soft boundary on hard examples, and mixup, that does not really differentiate hard and easy examples, have similar boundaries as claimed by the paper. Or is the similarity really of that high a degree?) (2) why extrapolations worked well on figures, but not on the dataset evaluated.

- The paper also proposed a Test-Time mixup idea, that simulates ensemble as it proposes several predictions that can be averaged. However, it is not exactly clear the advantage of this idea over vanilla mixup. It seems that works better on instances that were selected to correlate negatively to the test time performance (based on other models), is this a sign of good generalization or robustness? It is not quite clear. To validate whether the Test-Time mixup idea works, it would be better if more experiments on different datasets/backbone models are performed.


Questions:
1. How is Figure 2 drawn --- how are uncertainty measured for instances not belonging to the dataset.
2. A possible explanation for why the empirical experiment on ood in table 2 disagree with the plots for extrapolating: the ood samples that extrapolating has low confidence of might correspond to random images (pure noises), instead of actual images with different digits.



Typos:
* Section 3 last paragraph: where on trains a classifier -> where one trains a classifier
* In Table 2, for worst acc (ne), Ex-mixup-v2 has a higher value (0.509) than TT-mixup (0.507), and should be colored orange.

**Summary Of The Paper:**

This paper first analyzes the relationship between ensemble and mixup and empirically shows that both have some similarities in terms of their decision boundary --- there is some uncertainty region for mixup and ensemble, that is not seen in vanilla training. Then, the paper proposes a Test-Time mixup technique that allows mixup-trained models to simulate ensemble and generate multiple predictions for a single instance, which is then averaged. Finally, the paper talks about how mixup works in out of distribution generalization, and proposes to extrapolate the data instead of interpolating. Two methods are proposed, both extrapolating the input but for the labels one interpolating with an uniform distribution over all classes and another an one-hot on an inserted new dimension. The paper analyzed the mixup variants proposed and ensemble and showed that (1) the proposed Test-Time mixup works better on data points that is negatively correlated to test set performance and (2) the extrapolating methods do not work as well as original (interpolating) mixup.

**Summary Of The Review:**

This paper is overall interesting and brings some new insights to the understanding of mixup and deep ensemble. However, extra efforts may be required to justify its claims, which I believe can greatly improve the paper.

---

> ### Author Response · Authors · 2022-11-11
> **Rebuttal**
>
> Thank you so much for your thought-provoking review. Below, we try our best to shed some light into your questions.
>
> (1) Mixup and ensembles are similar because of the following reasoning. Ensembles average predictions of multiple zero-training-error ERMs, which are free to fluctuate randomly in-between training points. This leads to a linearization behavior on those regions. This is exactly the objective function that mixup minimizes. As seen in Figure 1, the mixup loss is implicitly minimized as we construct ensembles containing more members. On another level, both ensembles and mixup calibrate predictions better, since the corresponding machines have a much harder time obtaining saturated vectors. On the one hand, mixup training never produces one-hot vectors. On the other hand, the average of probabilities of different ERMs will also push far from one-hot vectors.
>
> (2) Regardless of experimental results, we believe our proposals and discussions regarding Ex-mixup are of interest and worth exploring further. We think extrapolation in high dimensions may not be as intuitive as Figure 3 due to results in https://arxiv.org/abs/2110.09485
>
> (3) We believe the main advantage of TT-mixup over mixup lays at OOD scenarios, where vanilla mixup methods have not observed data. Ensembling over multiple predictions cancels out high-confident single predictions, which leads to better uncertainty estimates and calibration outside of the training distribution.
>
> (Q1): Entropy of softmax prediction
> (Q2): That is an interesting point. We will add an OOD split consisting of random noise examples.

---

### Official Review · Reviewer_3JQa · 2022-10-29

**Confidence:** 4
**Correctness:** 3
**Technical Novelty And Significance:** 3
**Empirical Novelty And Significance:** 3
**Recommendation:** 8

**Clarity, Quality, Novelty And Reproducibility:**

As mentioned above, the paper is generally clear and precise in the writing, and I expect that it would be simple to replicate the experimental setup.

**Strength And Weaknesses:**

This paper has several strengths. At a high-level, the paper is clear and precise in the writing, with well-explained methods. Empirically, the experiments evaluated the methods on a reasonable set of metrics (accuracy, NLL ("data loss"), ECE, OOD accuracy), and all results included error bars, which is great. The takeaway from the paper is clear (use test-time mixup), and the paper also included a negative result (that the extrapolation mixup proposal wasn't generally beneficial), which is also refreshing to see.

In terms of weaknesses, it would be nice to better recognize that the proposed test-time mixup incurs a cost of K applications of the model at test time per example. Since both this test-time mixup and deep ensembles are trivially vectorized (not discussed in the paper), the main efficiency gain at test time is the potential for improved memory bandwidth due to having only one copy of the weights loaded in device memory. The factor of K predictions can still be prohibitively costly in terms of computation, and it would be great to better include that in the paper. Additionally, it could have been nice to see results for the combination of ensembles and mixup to better understand how they could complement each other, regardless of the increased inefficiency; this is a minor critique though.

**Summary Of The Paper:**

This paper proposes forming an "ensemble" at test time from a single mixup-trained model by averaging over multiple mixup predictions between the test example and K "reference" examples. They also experiment with improving the OOD uncertainty by training the mixup model with additional "extrapolation" examples in which they apply mixup between a training example and an extrapolation of that example, using the original label and an C+1 uncertainty label, with C being the number of classes. Empirically, they show that their original test-time mixup model (TT-mixup) outperforms even deep ensembles on the CIFAR-10 test set and CIFAR-10-Neg on accuracy, ECE, and diversity.

**Summary Of The Review:**

Overall, this is a well-written paper with a precise set of methods, a nice set of empirical results that demonstrate the effectiveness of the method against the gold-standard deep ensembles, and a clear takeaway (use test-time mixup). One main critique is that the test-time efficiency claim should be expanded upon to better reflect that the method is still quite costly at test time.


#### Minor
- p. 2: Near the end of the introduction, "ERM" is used before defining it in section 2. Though ERM is a common abbreviation, it's still nicer to define it upon first usage.
- p. 2: Capitalize "mixup" in the sentence beginning as "mixup ensembles are K times cheaper ..." in the Contributions section.
- p. 4, Table 1: What is "idem"?
- p. 5: MIMO also has the benefit of cheaper training and test step time than k applications of the model.
- p. 7: ECE original reference is Naeini et al, 2015.

---

> ### Author Response · Authors · 2022-11-11
> **Rebuttal**
>
> Thank you for your review. We will make sure to highlight the fact that TT-mixup requires K evaluations (as listed in Table 1).

---

### Official Review · Reviewer_AwxV · 2022-11-03

**Confidence:** 4
**Correctness:** 3
**Technical Novelty And Significance:** 3
**Empirical Novelty And Significance:** 3
**Recommendation:** 5

**Clarity, Quality, Novelty And Reproducibility:**

Clarity: Except a few small things, paper is clear.

Quality: Good

Novelty: The proposed TT-mixup and Ex-mixup is novel, but the impact of Ex-mixup is not clear. In the Figure:3 its shows positive impact, however, table-2 shows the negative impact.

Reproducibility: Hard (Proper data split, parameter and useful details are missing, also code is not available)

**Strength And Weaknesses:**

Strength:

1: The mixup ensemble is interesting, and it provides an efficient solution and shows a competitive result compared to the ERM ensemble.

2: The proposed TT-mixup and Ex-mixup are interesting, but the evaluation is somewhat weak.

3: The motivation is clear. The result in the Table-2 is comprehensive and Figure-3 seems interesting, especially the right figure for the Extra-mixup-v2.

Weaknesses:

1: The paper evaluated the model empirically. However, a proper theoretical justification can be provided for the ensemble kind of model. Is it possible to provide some bound/relation between the mixup and ensemble?

2: Most of the study, like generalization ability, OOD robustness, calibration, uncertainty estimates adversarial robustness are exist in the literature. So, in terms of these study the paper is not novel, but the proposed mixup is novel.

3: The results are evaluated only over a small dataset i.e. CIFAR10/CIFAR10-Neg. Which are not sufficient to evaluate the model efficacy. I request to the author to please provide the result for the ensemble and TT-mixup result for the CIFAR100 dataset. It contains a larger class and more similar classes.

4: In the section-4 Eq:2, term “u” is not clear. How it is used in the model? Please provide the details.

5: Reproducibility may be a key concern, mixup itself has a lot of randomness and here parameters and implementation details are not complete. Also, datasets split details are missing. Overall, reproducibility is hard. I request to the author to please provide the code with the details descriptions. What are OOD samples?.

6: In the Figure:4, on the X-axis there are number 1,2 what are that numbers?

7: Ex-mixup-v1 and Ex-mixup-v2 showing consistently poor result, when it will be useful? Did the author tried over some extrapolated data? In the Figure-3 (Extra-mixup-v2) it shows very impressive result but same pattern are not evident in the Table-2 why?



**Summary Of The Paper:**

The paper studied the ensemble perspective of the mixup. They proposed novel variant of mixup, i.e. TT-mixup and Ex-mixup. TT-mixup shows highly competitive result with the K-ensemble model, however taking K times less training and deployment cost. The experiment on the small data (CIFAR10) validates the efficacy of the proposed model.

**Summary Of The Review:**

Overall, the idea is interesting and applicable to many practical problems, but the evaluation is weak and theoretical justification is missing.
the contribution of the Ex-mixup is confusing.

Please refer to strength and weakness section for details.

---

> ### Author Response · Authors · 2022-11-11
> **Rebuttal**
>
> Thank you for your careful and insightful review. We address your raised weaknesses below:
>
> 1: One can show that the ensemble evaluation at $x$ tends to mixup as $lambda_k \to 1$. If the machine satisfies the mixup condition around $x$, then equation (1) holds.
>
> 2: Evaluation is performed across a variety of metrics and splits of CIFAR-based data, on using a unified codebase that we will release, with rigorous hyper-parameter search and model selection.
>
> 3: We are currently running experiments for CIFAR-100, and will be included in final version.
>
> 4: Stated in the lines below, $u = (1/C, ..., 1/C)$ is the maximum uncertainty label, where $C$ is the number of classes. For $C=2$, then $u = (1/2, 1/2)$.
>
> 5: We have uploaded the code, which produces the very same results under two different end-to-end runs. IID examples are CIFAR classes 1-5 (used for training); OOD examples are CIFAR classes 6-10.
>
> 6: Value of mixup loss at test examples.
>
> 7: Regardless of experimental results, we believe our proposals and discussions regarding Ex-mixup are of interest and worth exploring further. We think extrapolation in high dimensions may not be as intuitive as Figure 3 due to results in https://arxiv.org/abs/2110.09485

---

### Decision · Program_Chairs · 2023-01-20

**Decision:**

Reject

**Justification For Why Not Higher Score:**

N/A

**Justification For Why Not Lower Score:**

N/A

**Metareview: Summary, Strengths And Weaknesses:**

This paper presents an ensemble view on mixup in an attempt to explain its robustness and generalisation abilities. A mixup-ensemble with K members is formed by averaging predictions of a given (test) sample and K reference samples (test-time mixup). The authors have also proposed two variants of this approach to increase OOD accuracy and detection (called Ex-mixup).

While acknowledging that the proposed model is potentially useful, the reviewers raised several important concerns:
(1) more formal justifications are required to assess the scope and significance of this work contributions – see comments of reviewers AwxV, S9F4, 1zoz.
(2) writing and presentation clarity of the paper could be improved – see very detailed comments by Reviewer 1zoz;
(3) empirical results are mixed and can be improved – see all reviewers' comments.

Among these, (2) and (3) did not have a substantial impact on the decision, but would be helpful to address in a subsequent revision. However, (1) makes it very difficult to assess the benefits of the proposed approach, and was viewed by AC as a critical issue. We hope the reviews are useful for improving and revising the paper.